# Central cavity dehydration as a gating mechanism of potassium channels

Ruo-Xu Gu[1,2] & Bert L. de Groot ⬤[2] ✉

The hydrophobic gating model, in which ion permeation is inhibited by the hydrophobicity, rather than a physical occlusion of the nanopore, functions in various ion channels including potassium channels. Available research focused on the energy barriers for ion/water conduction due to the hydrophobicity, whereas how hydrophobic gating affects the function and structure of channels remains unclear. Here, we use potassium channels as examples and conduct molecular dynamics simulations to investigate this problem. Our simulations find channel activities (ion currents) highly correlated with cavity hydration level, implying insufficient hydration as a barrier for ion permeation. Enforced cavity dehydration successfully induces conformational transitions between known channel states, further implying cavity dewetting as a key step in the gating procedure of potassium channels utilizing different activation mechanisms. Our work reveals how the cavity dewetting is coupled to structural changes of potassium channels and how it affects channel activity. The conclusion may also apply to other ion channels.

Water and ion permeability across a pore of sub-nanometer size is a function of its radius, geometry, and hydrophilicity of the lining surface[1–4]. Waters confined in hydrophobic nanopores behave differently from the bulk, as characterized by stochastic liquid-vapor transition due to unfavorable water-hydrophobic surface interactions[3,4]. This phenomena leads to the concept of "hydrophobic gating", in which the conduction of waters and ions is inhibited without a physical occlusion of the nanopore[5]. Although first developed using simplified models and carbon nanotubes[3,6], the concept also applies to ion channels, whose permeation pathway is of nanometer size and the hydrophilicity of their surfaces is modified by amino acid side chains. One of the earliest tests of the concept on ion channels was performed by Corry for the acetylcholine receptor[7]. Afterwards, the "cavity dewetting" has been found to be a gating mechanism for various ion channels including other pentameric ligand gated ion channels (e.g., 5-HT$_3$ receptor[8]), mechanosensitive ion channels (e.g., Piezo[9], MscS[10], and MscL[11]), the CorA magnesium channel[12,13], and potassium ion channels[14], as reviewed comprehensively in Lynch et al.'s work[5]. The relationship between pore wettability and the pore radius and hydrophobicity has been investigated systematically for hundreds of ion channels in the protein

data bank by ref. 15, concluded with a model for hydrophobic gating prediction.

In this work, we focused on hydrophobic gating in potassium ion channels. Potassium channels are normally tetramers (or pseudo-tetramers) whose permeation pathway is surrounded by the pore domain (Fig. 1), in where each subunit contains two transmembrane (TM) helices connected by a pore helix and a conserved loop region[16]. The loop regions constitute the selectivity filter with four potassium binding sites at the extracellular half of the bilayers, beneath which is a central cavity formed by four inner TM helices[17] (see Fig. 1). The central cavity is also known as the activation gate, as conformational changes of the inner TM helices, coupled to motions of other domains responsible for biological signal sensing (e.g., the voltage sensing domain, calcium-binding domain), operate as a gating mechanism of potassium channels[18]. Bending and splaying of these TM helices open the permeation pathway[19,20]. In the traditional "bundle crossing" mechanism, the inner TM helices from the four subunits intersect in the closed state, resulting in a physically closed central cavity inaccessible to the intracellular side. This model was first proposed based on the crystal structure of KcsA[20], a pH-sensitive potassium channel from bacteria, and was also found for the cryo-EM

[1]School of Life Sciences and Biotechnology, Shanghai Jia Tong University, 800 Dongchuan Road, 200240 Shanghai, China. [2]Department of Theoretical and Computational Biophysics, Max-Planck Institute for Multidisciplinary Sciences, Am Fassberg 11, 37077 Göttingen, Germany. ✉e-mail: bgroot@gwdg.de

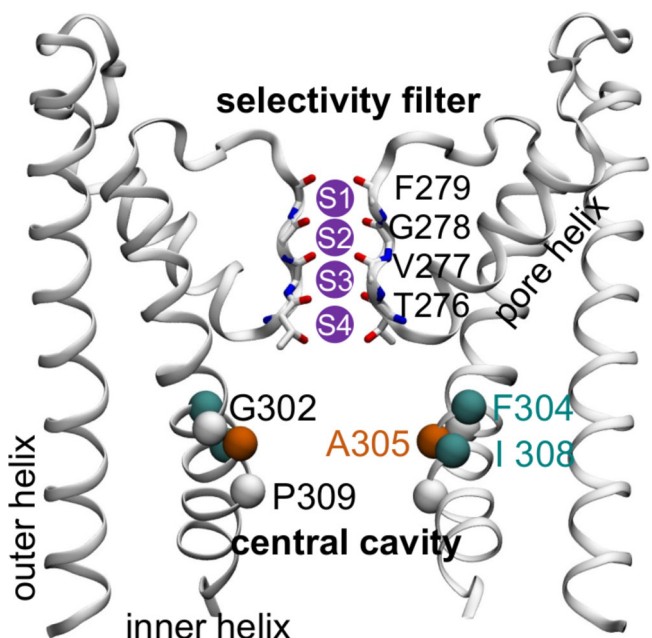

**Fig. 1 | Structure of the pore domain of Aplysia BK channel.** Two subunits of the tetramer are shown for clarity. The four potassium binding sites at the selectivity filter are labeled as S1–S4. The backbone of the residues constituting the selectivity filter, as well as the Cα atoms of several residues from the inner helices critical for gating and ion conduction, are labeled.

structure of MthK[21], a calcium gated channel from Methanothermobacter thermautotrophicus.

However, this model is not appliable to some potassium channels, as suggested by both structural biological data and biochemical experiments. One example might be a calcium-gated, large conductance potassium channel (the Aplysia BK channel), whose cryo-EM structures showed a central cavity accessible to the cytoplasmic side in both the open and closed states[22,23]. Channel inhibition by quaternary ammonium (QA), channel blockers binding to the central cavity, is not state-dependent for BK[24], also implying accessibility of the cavity to QA in the closed state, contradicting a physically closed entrance predicted by the bundle crossing model. Among the attempts looking for the activation gate of BK, ref. 25 proposed the "physically open, functionally closed" hydrophobic gating model based on molecular dynamics (MD) simulations, successfully reconciled the seemed contradictory experimental results. Similar mechanisms may also apply for other potassium channels. For instance, existing evidence suggested against the bundle crossing mechanism for the calcium gated, small conductance potassium channels (SK[26,27]), as well as some of the two pore domain potassium channels (K2P[16]). The hydrophobicity deep in the central cavity of TWIK-1, a K2P channel, resulted an energy barrier for ion conduction, which was responsible for its low ion permeability in physiological conditions[28]. Moreover, cavity dewetting also plays a role even in the channels utilizing the bundle crossing mechanism. A reverse voltage across the bilayer prompted a hydrophobic collapse of the central cavity for Kv1.2 channel[29,30], which was believed to be correlated with low or no inward conduction and outward rectify under negative voltages for some of the potassium channels (e.g., KcsA[31]). Besides, the dewetting of the central cavity explained the sensitivity to osmotic pressure for potassium channels such as KcsA, which argues that hyperosmotic conditions reduce the open probability and ion currents[32]. However, available studies regarding hydrophobic gating in ion channels focuses on the energy barrier blocking ion and water permeation due to the hydrophobicity, while how the cavity dewetting and protein conformational transition are coupled remains unclear.

In this work, we use potassium channels as examples and conduct MD simulations to provide insight into this question. Specifically, we select two channels, the Aplysia BK channel (aBK[22]), which is believed to utilize hydrophobic gating to operate ion conduction, and its homolog MthK[21], whose cryo-EM structure suggested a bundle crossing mechanism. We first investigate the ion permeability of aBK as a function of its cavity hydration level in both the open and closed states to quantify the effects of hydrophobic gating on protein function. We then perform non-equilibrium simulations using different methods to: (a) induce conformational transition from open to closed state by enforced cavity dewetting and (b) explore the cavity dewetting process as a consequence of conformational transition. In conclusion, we not only establish the relationship between protein activity and cavity hydration level, but also reveal the coupling mechanism between protein conformational transition and cavity dewetting. Simulations of MthK, which employs an activation gating mechanism different from aBK, show consistent results. Our findings may also apply for other members of the potassium channel family, or other ion channels utilizing similar activation mechanism.

## Results

### Ion currents for the two states of BK

In order to compare the ion permeability of different structures revealed by cryo-EM, we conducted MD simulations of the pore domain of the BK channel in its open and closed states (Fig. 2a). The ion currents of -14.4 and 1.3 pA, obtained with a transmembrane voltage of 300 mV and a [K⁺] of 1 M, are consistent with the channel states inferred from experiments.

### Different cavity hydration levels of the two states

A much higher hydration degree was identified for the open state, as characterized by the average number of water molecules in the cavity (~61 and 23 for the two states, respectively, Fig. 2e). Further analyses of the trajectories indicated different hydration degrees to the different conformations of the inner TM helices in these two states, as shown in Fig. 2a. Specifically, a kink around Gly302 (Fig. 2b) in the open state resulted a larger bending of the TM helices and an open cavity, whereas lacking the kink not only made the TM helices straighter but also shifted the side chain of some hydrophobic residues (e.g., Phe304 and Ile308) toward the central cavity (Fig. 2a–d), which in turn induced dehydration. We note that the kink in the open state is stabilized by a water molecular binding to the backbone of the TM helices (Fig. 2a). We used the backbone hydrogen bonds around Gly302 (Fig. 2b, d) and the Phe304 side chain orientation (Fig. 2c, d), to characterize the abovementioned structural differences between two states, respectively. Moreover, we calculated the free energy profiles of water molecules and potassium ions entering the cavity (Fig. 2g). As expected, a higher energy barrier for water (~0.8 v.s. 2 $k_BT$) and potassium (~2 v.s. 7 $k_BT$) were found for the dehydrated closed state and the positions of the energy barriers were located in the vicinity of Phe304 and Ile308.

### Cavity hydration degree correlates with channel function

To further evaluate the effects of cavity dehydration on ion permeation, we introduced mutations to both of the open and closed structures, with the aim to modulate water distributions in the cavity. We first tried to increase the hydrophobicity of the central cavity in the open state by mutation of Ala305, whose side chain points toward the cavity (Fig. 2a), to hydrophobic residues of different sizes such as Val and Leu. These mutations reduced the cavity hydration level to different extents (water numbers of ~40 and ~18, respectively, compared to ~61 for wild type, Fig. 2e) and resulted in smaller ion currents (~6.8 and 0.3 pA, compared to -14.4 pA of wild type, Fig. 2e). As a comparison, the Ala305Glu mutant, whose large side chains also occupies the space in the central cavity, maintained hydration degree

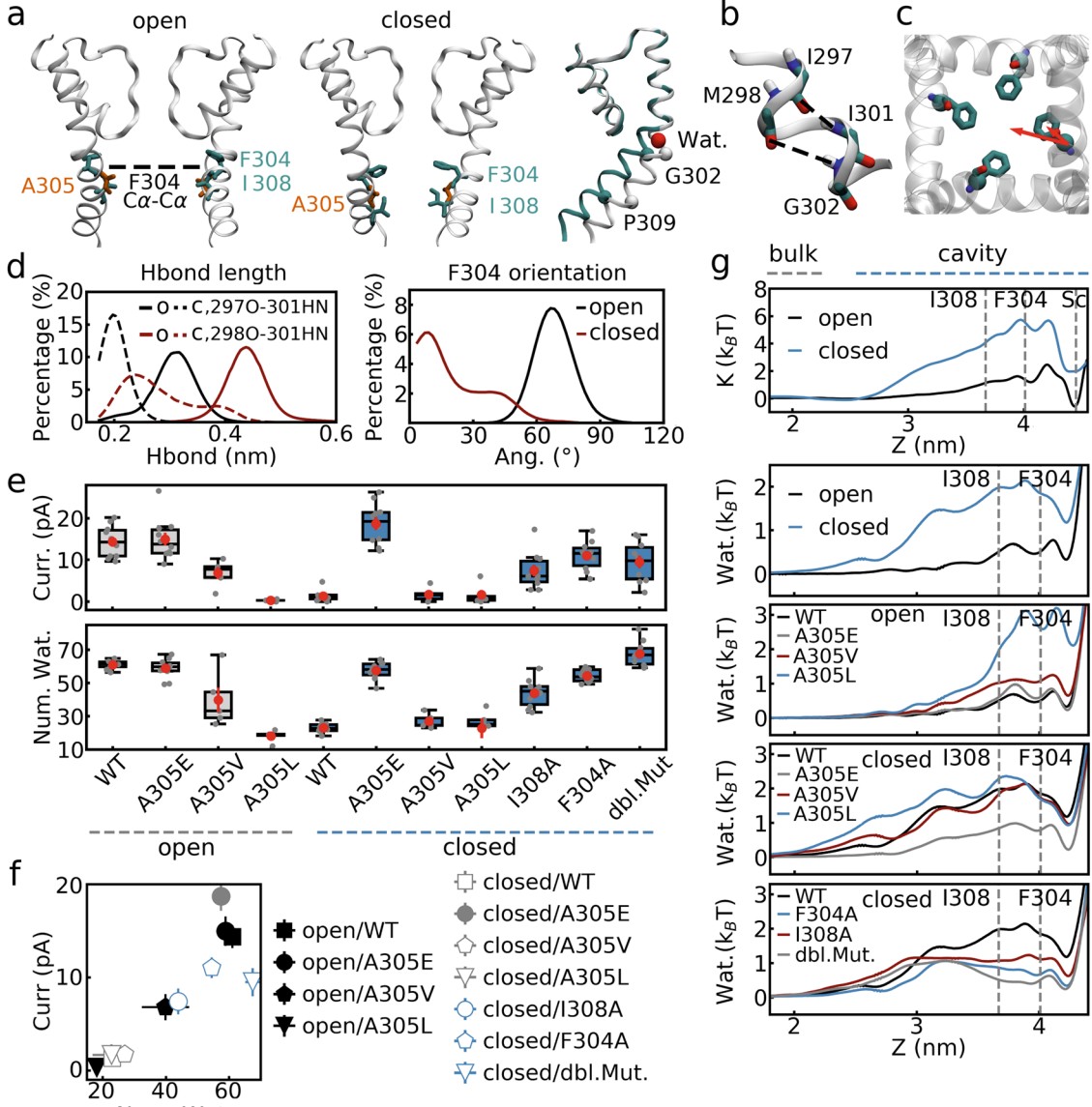

**Fig. 2 | Ion permeability is highly correlated with central cavity hydration level.**
**a** Conformations of the open and closed states of the BK channel, as well as superimposition of their structures (open: cyan; closed: white). Only two subunits are shown for clarity. The sidechains of F304, I308, and A305 of the inner helix, which were mutated in this work, are shown in stick model in cyan and orange. Cα atoms of G302 and P309, as well as the water molecule (red) stabilizing the kink, are shown in space-filling model. **b** Backbone hydrogen bonds at the kink region of the inner helix. Possible hydrogen bonds between I297 and I301, and the one between M298 and G302 are marked by dashed lines. **c** The angle defined by the red arrows is used to describe the F304 side-chain orientation. **d** Distributions of the hydrogen bonds at the kink and the F304 side-chain orientation in the open (O) and closed (C) states. **e** Ion currents and numbers of waters in the central cavity. The data are presented as boxplot. The center line, box limits, and whiskers represent the

median, upper and lower quartiles, and 1.5× interquartile range. The data points are shown in gray dots, while the averages and errors (standard error of the mean, s.e.m.) are shown in red. "dbl Mut." standards for the F304A/I308A double mutant. **f** Correlation between ion currents and numbers of waters in the central cavity. The error bar reported is the standard error of the mean (s.e.m.). **g** Free energy profiles of potassium ions entering into the central cavity in the simulations of open and closed states of the wild-type channel, and the corresponding profiles for waters in the simulations of wild-type channel and mutants. Positions of F304 and I308, as well as the position of the Sc potassium binding site, are marked by gray dashed lines. $n = 5$ independent simulation replicas were performed for open/A305V, open/A305L, closed/A305V, and closed/A305L systems, whereas $n = 10$ independent simulation replicas were conducted for the other systems.

and ion conduction rate similar to the wild type due to the hydrophilicity of glutamic acid (Fig. 2e).

As expected, using the closed state as the initial structure, the Ala305Val and Ala305Leu mutants remained dewetted, while the hydrophilic side chains of Ala305Glu mutant hydrated the central cavity to the level of the open state (water number of ~57) and resulted in a relatively high ion current (~18.7 pA, Fig. 2e). As mentioned above, Phe304 and Ile308 occluded the central cavity and are inferred to be responsible for cavity dehydration in the closed state (Fig. 2a). In this regard, we decreased the hydrophobicity and

increased the size of the central cavity by mutating these two residues to Ala, respectively. Mutation of either residue was able to hydrate the cavity, with the Phe304Ala mutation showing a slightly higher hydration level due to its larger side chain (water numbers of ~44 and 54, respectively), suggesting both residues are critical for the channel inhibition in the closed state. The double mutation resulted the highest hydration degree as expected (water number of ~68) (Fig. 2e). Notably, high ion currents (~7.4, 11.0, and 9.5 pA) were found for these mutants as compared to the wild-type (Fig. 2e) in the closed state.

The free energy profiles for water molecules and potassium ions entering into the cavity of these mutants were compared to the results of the wild type (Fig. 2g, Supplementary Fig. 4). The results are consistent with the trends of the values of ion current and hydration level.

Most of the abovementioned mutations did not change the initial structures for the backbone during simulations (see Supplementary Fig. 5, in this case, changes in hydration levels were mainly due to sidechains), except for the Ala305Glu mutant of the closed state, in which a conformational transition from the closed to the open state was observed (see "cavity hydration induced channel opening" section for details). Marginal structural changes observed for the Ala305Val and Ala305Leu mutants of the open state were ascribed to a lateral contraction of the four inner helices induced by the attraction between the hydrophobic sidechains, rather than a conformational transition between two states (Supplementary Fig. 5). The other mutations had little effect on the protein structure. Thus, we have five simulations in total with an open-state backbone structure and different hydration levels. A strong relationship between ion currents and cavity hydration levels was identified for these simulations: fully hydrated simulations (open/WT, open/Ala305Glu, closed/Ala305Glu) showed the largest conduction rate, medium hydration level (open/Ala305Val) resulted in a significantly decreased current, whereas ion conduction was inhibited if the hydration level was as low as the closed state of the wild type channel (open/Ala305Leu, see the closed symbols in Fig. 2f). Similar results were found for the simulations with the closed state backbone structure: almost no ion conduction in the cases of dwetted cavity (closed/WT, closed/Ala305Val, closed/Ala305Leu) and significantly larger currents in the cases of medium (closed/Ile308Ala, closed/Phe304Ala) and high (closed/double mutation) hydration levels (see the open symbols in Fig. 2f). We note that the cavity of the double mutant of the closed state had as many water molecules as the fully hydrated cases of the open state but showed a smaller ion current (-9.5 v.s. 14–18 pA). The different solvation degrees of potassium ions in the cavities between them were identified as a possible reason (Supplementary Fig. 6, Supplementary Table 4, see discussion). We also hypothesize that the different cavity structures (water density maps in Supplementary Fig. 7 showed a tubular shape structure of the cavity for the former and a funnel shape structure for the latter) is another reason for this difference.

We note that inhibition of ion conduction did not require complete dehydration, as there were significant numbers of waters in the cavity of the non-conducting states (ion currents of -1 pA or lower, Fig. 2e, the water densities in the cavity of the non-conducting states were about -0.4–0.8 times of the bulk value, see Supplementary Fig. 7). We conclude that, in this BK channel, the degree of cavity hydration is critical for the ion conduction rate and the different hydration levels of the two wild type states are responsible for their different ion permeability.

The above conclusions are not affected by the high transmembrane voltage and ion concentration used in this work, as suggested by control simulations under different combinations of voltages and ion concentrations (1 M/150 mV, 0.15 M/300 mV, and 0.15 M/150 mV) (Supplementary Fig. 8, Supplementary Table 1, Supplementary Note 1). The same correlation between cavity hydration level and ion currents was found under 0.15 M/300 mV and 0.15 M/150 mV (Supplementary Fig. 8d). Lower voltages and ion concentrations did not affect the number of water molecules, but reduced the number of potassium ions in the cavities (Supplementary Fig. 8f, g). The currents are more sensitive to the voltages than the ion concentrations in the tested cases (Supplementary Note 1).

We measured the fluctuation of the cavity hydration to further explore the hydrophobic gating model in the case of potassium channels. Relative standard deviations (ratio of standard deviation to the mean) suggested much higher fluctuations of the cavity hydration for the less hydrated cases (-0.1, -0.2, and -0.3 for the fully hydrated,

medium hydrated and dewetted cavities, see Supplementary Fig. 10). Taken the dewetted cavity of the closed state of the wild type channel as an example: the cavity contained -23 water molecules on average with fast fluctuation between 10 and 30 in tens of nanoseconds. Trajectories of single water molecules showed that they usually stayed in the cavity for less than 10 ns (Supplementary Fig. 10).

## Cavity dewetted quickly in MD simulations of the closed state of BK

To further illustrate the structural elements responsible for cavity dehydration and inhibition of ion permeability, we analyzed the dewetting procedure and the channel structures in detail in the MD simulations of the closed state.

In the initial conformation of our closed state simulations, the central cavity was empty. We then conducted 0.25 μs equilibrium simulations in which the protein backbone was restrained for the first 0.05 μs. Water molecules entered and the cavity was filled with -80 water molecules during the backbone restrained simulations (Fig. 3a, b). However, the cavity dewetted within 0.1 μs after the restraints were removed, as shown in Fig. 3b.

The abovementioned cavity dewetting was ascribed to conformational adjustment of the protein, which involved a compaction of the four inner TM helices, and a shift of Phe304 side chains as well, as characterized by the Phe304 Cα-Cα distances of the opposite subunits and the Phe304 side chain angle in Fig. 3b. It seemed that Phe304 residues shift their side chains toward the central cavity and the attraction between the hydrophobic side chains induced further compaction of the protein backbone. Superimposition of the average structure in MD simulations to the cryo-EM structure (Fig. 3c, d) showed that the main conformational differences located at the inner TM helices, from the kink region to the C-terminus, implying a possible rigid body motion hinging at the kink point. The conformational change was moderate, as indicated by the RMSD value of 0.23 nm. Note that the quick dehydration and the conformational changes we found here is similar to ref. 25 simulations of the human BK channel.

In simulations with the backbone of the inner helices restrained and a transmembrane voltage of 300 mV, the presumable closed cryo-EM structure was as conductive as the open state (13.3 v.s. 14.4 pA, see Supplementary Table 1). In these simulations, the fully hydrated cavity, which was ascribed to the wide splaying of the four TM helices and the orientation of the Phe304 sidechains (pointing to the subunit interfaces), allowed ion permeation (Fig. 3). This observation highlighted the role of cavity dewetting in the ion conduction inhibition. This hydrated experimental structure and the relaxed dewetted conformation were referred as "closed cryo-EM structure" and "closed state" in this work.

## Cavity dewetting induced conformational transition

The conformational transition from the open to the closed state and cavity dewetting should be coupled with each other if dehydration works as a gating mechanism. In this regard, we conducted non-equilibrium simulations to explore the correlation between cavity dehydration and the protein conformation.

Specifically, we started from the open state, pulled waters out of the cavity to enforce dehydration and analyzed how the protein changed its conformation during this process. The simulations were repeated multiple times. In each replica of 1.5 μs, the waters were pulled out of the cavity during the first 0.5 μs, followed by another 0.5 μs simulations with the cavity restrained to be empty. The system was then equilibrated for the last 0.5 μs without any restraints.

A conformational transition from the open to the closed state happened for 13 out of the 17 replicas (Supplementary Table 1, success rate of -75%), while for the remaining ones, either forced dehydration was not successful or partial unwinding of the TM helices was observed. The closed structures obtained by forced dewetting

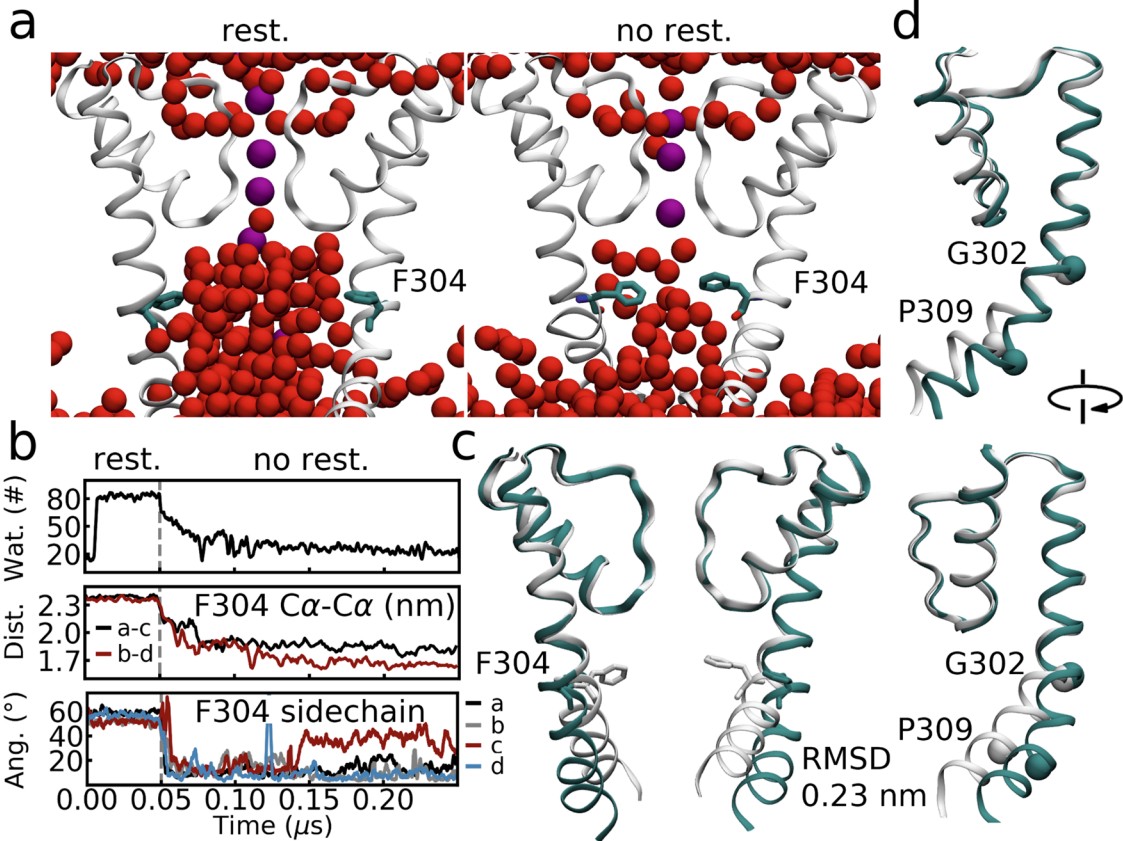

**Fig. 3 | Central cavity dehydrated quickly in MD simulations of the closed BK channel. a** Cavity hydration levels of the closed state in restrained and unrestrained simulations. The F304 side chains are shown in cyan. The water oxygen atoms and potassium ions are shown in space filling model in red and purple. **b** The number of water molecules in cavity, F304 Cα-Cα distance, and F304 side chain orientation as a function of simulation time. Note that the protein backbone is restrained in the first 0.05 μs of the simulations. **c**, **d** Alignment of the cryo-EM structure (cyan) and the average structure (white) in MD simulations. Only the pore helix, the selectivity filter and the inner helix are shown for clarity. F304 side chains, and the Cα atoms of G302 and P309 are shown in stick model and space filling model.

simulations, as well as their hydration levels (Fig. 4a–c, also see Supplementary Fig. 13), were similar to the equilibrium simulations of the corresponding state, as indicated by the boxplot of RMSD values of multiple replicas (average value of ~0.24 nm, Fig. 4b) and the number of water molecules in the cavity (average value of ~14 v.s. ~23, Fig. 4c). The backbone hydrogen bonds at the kink region re-formed for most subunits in the non-equilibrium simulations (Fig. 4d), indicating that the key differences between the open and closed states were able to be modulated by cavity dehydration. Moreover, the closed structure averaged over all of the non-equilibrium replicas was aligned to the average structure of the equilibrium simulations of the closed state, as shown in Fig. 4a, and a RMSD of 0.12 nm indicated very high similarity between them.

Consistent with previous analyses and hypotheses, the conformational transition procedure involved rewinding of the kink, compaction of the four inner TM helices, and changes of the Phe304 side chain orientation as well. Taken one representative simulation as an example (Fig. 4e, f), the central cavity dehydrated quickly under external restraints (see water numbers in Fig. 4e), while the formation of backbone hydrogen bonds at the kink, compaction of inner TM helices (characterized by Phe304 Cα-Cα distance), and shifting of the Phe304 side chains (characterized by side chain angle) happened gradually afterwards in ~0.5 μs (Fig. 4e). The protein conformation deviated from the open state and converged to the closed state during this process (as suggested by RMSD in Fig. 4e), and remained closed and dehydrated (Fig. 4b–e) after the restraints were removed. Unbending and compaction of the inner TM helices are also shown by the snapshots at key time points of this procedure in Fig. 4f.

Distributions of hydrogen bonds at the kink region of the closed structure in this representative simulation also closely resemble those of the equilibrium simulations of the closed state (Fig. 4d), implying, again, highly similar conformations.

These data suggest that cavity dewetting successfully produced a closed state from an open conformation and the resulting structure was similar to those in the equilibrium simulations of the closed state, highlighting the role of dehydration in conformational transition and channel gating.

The kink of the inner TM helices was ascribed to a glycine (Gly302), which is known to disrupt the conformation of α-helices. Besides, Pro309 also plays a role in bending the inner TM helices and affects the conformation of the closed state (see Supplementary Note 2, Supplementary Figs. 11, 12). In this regard, we conducted equilibrium simulations of the Gly302Ala and Pro309Ala mutants in their closed states and the abovementioned non-equilibrium simulations starting from their open states to study the effects of these two mutations on channel gating.

As expected, for Gly302Ala mutants, the enforced dewetting simulations induced conformational transitions with much better convergence than the wild-type simulations, suggesting that the Gly302Ala mutation favors formation of the closed state as compared to the wild type (see Supplementary Note 3, Fig. 4b, c, Supplementary Figs. 14–17). For Pro309Ala mutants, straightening of the inner TM helices due to the mutation affected the tetramer structure arrangement in both of the equilibrium and non-equilibrium simulations (see Supplementary Note 3, Fig. 4b, c, Supplementary Figs. 14–17). We conclude that bending of the inner TM helices around Pro309 is

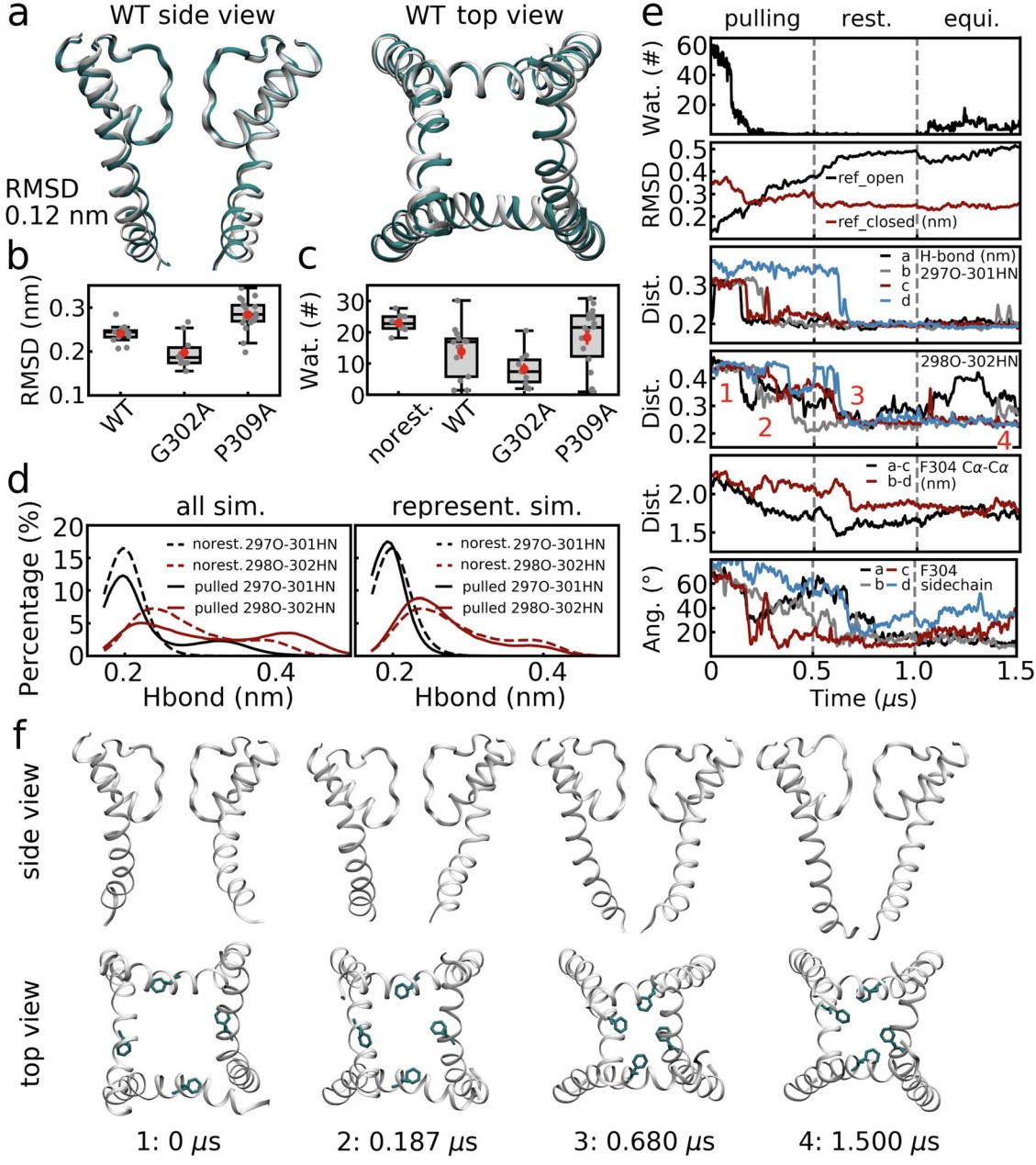

**Fig. 4 | Enforced cavity dehydration induced conformational transition from the open to closed state.** Comparison of the closed conformation (wild type) in non-equilibrium and equilibrium simulations. **a** The average structure for all of the enforced dehydration simulations (cyan) is aligned to that of the equilibrium simulations of the closed state (white). **b** Boxplot of the RMSD values. The RMSD of the average structure for each of the non-equilibrium simulation was calculated relative to the average structure of corresponding equilibrium simulations. **c** Boxplot of the numbers of water molecules in the cavity in non-equilibrium simulations. Results of the equilibrium simulations of the wild type ("no rest.") are also shown for comparison. In (**b**, **c**), results of the wild type ($n = 13$) and the G302A ($n = 10$), P309A ($n = 19$) mutants are shown. The center line, box limits, and whiskers represent the median, upper and lower quartiles, and $1.5 \times$ interquartile range. The data points are shown in gray dots, while the averages and errors (standard error of the mean, s.e.m.) are shown in red. **d** Length distributions of the hydrogen bonds at

the kink of the wild-type channel. Results of all non-equilibrium simulation replicas (left) and a representative case (right) were compared to the corresponding results of the equilibrium simulations (dashed lines). **e** A representative enforced dewetting simulation. Number of water molecules in the cavity, RMSD values relative to the open and closed states, hydrogen bond length (2970-301HN, and 2980-302HN), F304 Cα-Cα distance, and F304 side chain orientation are shown as a function of simulation time. Note that waters were pulled in the first 0.5 μs ("pulling"), the cavity was restrained to be empty in the following 0.5 μs ("rest."), and no restraints applied for the last 0.5 μs ("equi."). The lines were smoothed for clarity by averaging every 5 or 10 data points. **f** Snapshots of the BK channel at the key time points in the representative simulation to shown the procedure of its conformational transition. These time points are labeled as 1–4 in (**e**). Protein backbones and F304 sidechains are shown in white and cyan.

essential for proper arrangement of the tetramer structure by avoiding steric hindrance between the four subunits.

## Conformational transition resulted in cavity dehydration

The above simulations induced conformational transitions to happen by the restraining cavity hydration level. One may wonder about the

reverse process, i.e., how the cavity hydration level changes during the process of conformational transition. We further explored the coupling between cavity dehydration and protein conformational transition by directly modulating the conformation of the inner TM helices.

In equilibrium simulations, the kink in the open state was stabilized by a water molecule hydrogen bonding with the protein

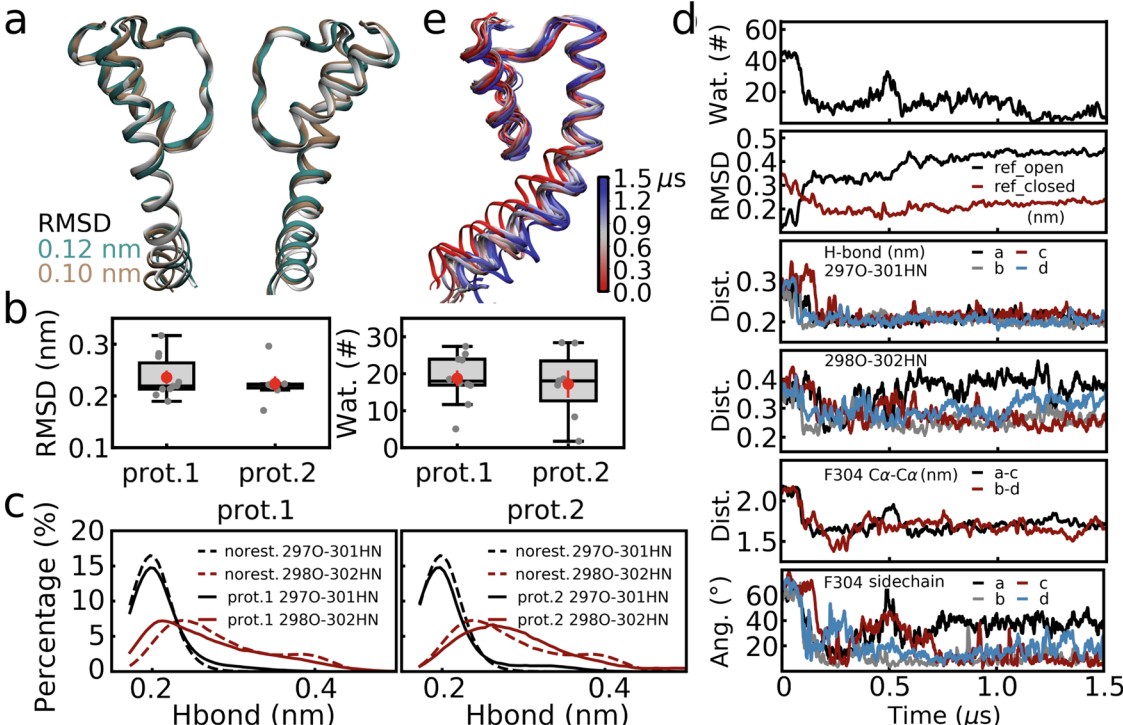

**Fig. 5 | Conformational transition from the open to closed state resulted in cavity dehydration. a** Comparison of the conformation of the closed state channel obtained from non-equilibrium (protocol 1: cyan; protocol 2: brown) and equilibrium simulations (white). The average structures of all simulation replicas using dummy atoms are aligned to that of the equilibrium simulations. **b** Boxplot of RMSD values (left) and the numbers of waters in the cavity for each simulation replica. Similar to Fig. 4, the RMSD of the average structure for each non-equilibrium simulation replica was calculated relative to that for the equilibrium simulations. $n = 10$ and $n = 7$ independent simulation replicas were conducted for protocol 1 and protocol 2. The center line, box limits, and whiskers represent the median, upper and lower quartiles, and 1.5 × interquartile range. Data points are shown in gray dots, whereas the averages and errors (s.e.m.) are shown in red. **c** Conformation of the inner helices described by the length distributions of the hydrogen bonds at the kink. Results for all simulation replicas were shown separately for protocol 1 (left) and 2 (right). Results of equilibrium simulations are also plotted for comparison. **d** A representative simulation using protocol 1. Water number in the cavity, RMSD values relative to the open and closed states, hydrogen bond length (297O-301HN, and 298O-302HN), F304 Cα-Cα distance, and F304 side chain orientation are shown as a function of simulation time. The lines were smoothed for clarity by averaging every 5 or 10 data points. **e** Structural alignment of snapshots of the representative simulation in (**d**) to show the conformational transition procedure.

backbone. In this regard, we placed a dummy atom around the kink which only interacted with water oxygen atoms, to repel the waters and prompt the reformation of backbone hydrogen bonds. In multiple parallel simulations of different lengths, we observed conformational transition from open to closed states, which in turn resulted in cavity dehydration. As in the enforced dewetting simulations, RMSD values of the average structure of each replica relative to the average structure of equilibrium simulations (~0.24 nm, Fig. 5b), the corresponding value of the average structure over all replicas (0.12 nm, Fig. 5a) and distributions of hydrogen bonds at the kink region (Fig. 5c) indicated a closed structure similar to that in equilibrium simulations starting from the closed state. The average number of water molecules in the cavity in Fig. 5b also showed a comparable hydration level.

A conformational transition mechanism similar to the enforced dewetting simulations was found, as shown by a representative simulation in Fig. 5d-e. A re-formation of backbone hydrogen bonds, compaction of four TM helices (Phe304 Cα-Cα distance), as well as a shift of the Phe304 side chain (Phe304 angle), which were used to characterize the protein conformation, happened quickly in ~0.2 μs at the beginning of the simulation, while the cavity was dehydrated almost simultaneously. Snapshots showing the cavity hydration level changes during the protein conformational transition are presented in Supplementary Fig. 18. The RMSD (Fig. 5d) relative to the open and closed states indicated convergence to the closed state during this process. Superimposition of the snapshots from this simulation (Fig. 5e) implied that the main conformational changes happened at the kink region.

In the above simulations (addressed as protocol 1), partial backbone unwinding around Val294, one helical turn ahead of the kink, was found in some of the simulation replicas (Supplementary Fig. 19). To investigate this issue, we restrained the helical hydrogen bonds (Supplementary Table 2) and redid the simulations (addressed as protocol 2). In this group of simulations, we obtained closed structures of similar hydration levels, but with much better convergence than the simulations without restraining these hydrogen bonds (as shown by the distribution of RMSD in Fig. 5b and a reduced RMSD value of 0.10 nm to the closed state, Fig. 5a). A representative simulation is shown in Supplementary Fig. 20.

Based on the above data, we conclude that repelling of the water molecule critical for kink formation was able to change the conformation from the open to the closed state, which finally induced cavity dehydration. These data, again, highlighted coupling between cavity dehydration and channel gating.

### Cavity hydration-induced channel opening

We also explored whether hydration of the central cavity can induce a conformational transition from the closed state to the open state. Cavity hydrations induced by either mutation (Ala305Glu, whose results regarding currents were also mentioned above) or restraining water molecules in the cavity were able to drive the conformational transition from the closed state toward the open state, reinforcing the idea of coupling between cavity hydration level and protein conformation. Our results implied that cavity hydration worked as a driving force for the conformational transition from the closed to the

open state, although it may not be able to induce complete a conformational transition by itself, as suggested by the asymmetrical conformational behaviors of the four subunits in our simulations (See Supplementary Note 4 and Supplementary Figs. 21, 22 for details).

### An intermediate state of MthK induced by cavity dewetting

Different from the BK channels, the closed state of MthK suggested a bundle crossing mechanism in which the central cavity is sealed by the intersection of the TM helices[21]. Therefore, we took it as another example to explore the role of cavity dehydration in channel gating. We conducted the abovementioned two non-equilibrium simulation protocols of MthK and obtained a conformation in which the cavity was dewetted but not sealed by helical intersection. In this conformation, the cavity kept dewetted due to the Phe87 sidechains (corresponding to Phe304 in aBK) and the bending angles of the inner TM helices fluctuated in a wide range. These data also implied the significance of cavity dewetting in conformational transitions in potassium channels, although what we sampled was likely to be an intermediate state (see Supplementary Note 5, Supplementary Figs. 23–25 for more details).

### Lipid–protein interactions

Lipid distributions in the cavity were observed in some simulation replicas for both the open and closed states of aBK. In both cases, lipids entered into the cavity via a crack between the subunit interfaces, which was enlarged occasionally due to fluctuations of the channel structures. In the open state, lipids distributed in the cavity might dehydrate the channel and inhibit ion permeation, whereas lipid–protein interactions often disrupted the tetramer structure of the channel in the closed state. We note that not all of our simulations had lipid distributions in the cavity. Whether the lipid-protein interactions observed in this work is physiologically relevant is debatable (see discussion), although lipid-protein interactions have been proposed to be involved in channel gating[16,21,25,33,34]. We used dummy atoms (see Supplementary Methods) to prevent lipids from entering into the cavity and to maintain the tetramer structure of the channel, so that we can focus on cavity dewetting and achieve better sampling statistics for the above-discussed simulations. Comparison of the simulations using dummy atoms and those without dummy atoms but lipid distributions in the cavity were not found indicated little differences for ion conduction rate and cavity hydration level. We therefore argue that the main conclusions of this work are not affected by the dummy atoms (see Supplementary Note 6, Supplementary Figs. 26, 27 for details, also see the discussion).

## Discussion

We conducted molecular dynamics simulations to investigate cavity dewetting as a gating mechanism of potassium channels. We first studied BK, a potassium channel which presumably utilizes hydrophobic gating to operate ion conduction. Equilibrium simulations revealed channel conductance highly correlated with the cavity hydration level, which highlights the role of water in the cavity for ion conduction and implies the existence of a hydrophobic gate. Non-equilibrium simulations successfully sampled conformational transition from the open to the closed state, and revealed the coupling between protein structural changes and the cavity dewetting, which further confirmed the role of cavity hydration on protein structure and function. We also studied MthK, a potassium channel which may employ bundle crossing as its gating mechanism. Although we did not obtain a stable closed state with a sealed central cavity, we sampled a conformation with the four TM helices splayed moderately and the central cavity dewetted, which is likely an intermediate structure of the protein conformational transition. Our work proved central cavity dehydration as a key mechanism of channel gating and function, for both proteins which may or may not utilize hydrophobic gating to modulate ion conduction.

Jia et al.[25] studied the existence of hydrophobic gating in human BK (hBK) by exploring cavity dewetting and comparing the free energy profiles for potassium and quaternary ammoniums entering into the cavity using simulations without applying transmembrane voltage. In this work, we further investigated how the hydrophobic gate inhibits channel function under transmembrane voltage. We modulated the central cavity hydration levels of Aplysia BK (aBK) to different extents by mutating residues lining the cavity surface. The resulting ion currents were found to be highly correlated with the number of water molecules in the central cavity (Fig. 2f), implying an energy barrier for ion permeation due to insufficient cavity hydration in simulations with fewer water molecules in the cavity. However, it was not essential for the cavity to be completely dry to inhibit ion conduction, as the cavity maintained a certain degree of hydration in the simulations with no or very low ion permeability (Fig. 2e, f, Supplementary Table 1). Actually, the average water density in the cavity of the closed state was -0.4−0.8 times of the bulk value (Supplementary Fig. 7). These data are consistent with the "physically open, functionally closed" hydrophobic gating model. Moreover, our data may also provide a possible explanation for different ion conduction rate of different channels:[19] different cavity opening degrees and hydration levels create different energy barriers for ion permeation.

We calculated the number of water molecules within the first and the second solvation shells of potassium ions in the cavity (Supplementary Table 4, Supplementary Fig. 6). The first solvation shells remained unaffected. However, the second solvation shells in the cavities of different hydration levels were all perturbed compared to the results in bulk. The average number of water molecules within the first two solvation shells (Supplementary Table 4) were smaller and the corresponding distributions (Supplementary Fig. 6) were shifted to the left in cavities with lower hydration levels than in those with higher hydration levels. The reduced solvation degrees of potassium ions provide possible explanations for the reduced currents in simulations with more dewetted cavities. Note that the Ala305Glu simulations were outliers: the solvation degrees were lower than simulations with similar cavity hydration levels, probably due to interactions between negatively charged residues and potassium ions. The different solvation degrees may also explain the situation of the double mutation: although it showed the highest cavity hydration level, the distribution of the ion solvation degree was shifted to the left compared to the open-state wild-type channel. The distribution was similar to that of the Phe304Ala mutant, whose cavity was hydrated at a medium level. Although the average number of water molecules in the first two solvation shells of the double mutant and Phe304Ala mutant did not show statistical differences with that of the wild-type channel in the open state, the shifts of the distributions cannot be ignored. We propose that the shape of the cavity may play a role: the ion solvation shell was easier to be perturbed in the narrower and longer cavity of the closed state than in the wider and shorter cavity of the open state (Supplementary Fig. 6, 7).

Key residues critical for cavity hydration levels have been identified. Jia et al.'s simulations of hBK[25] suggested that Phe315, Val319, and Ile323 (equivalents of Phe304, Ile308, and Ala312 in aBK) pointed toward the central cavity and at least some of them may be responsible for the depletion of water molecules from the cavity. Similarly, cavity dewetting in aBK was ascribed to Phe304 and Ile308. Mutation of any of these two residues was found to increase cavity hydration significantly (Fig. 2a, e). However, the side chain of Ala312 in aBK was too small to play a role, and the Ala312Val mutants did not reduce the cavity hydration level further in our simulations (number of water molecules of -27, Supplementary Table 1). The mutation of Ala316 to the charged Asp in hBK (equivalent of Ala305 in aBK) led to a constitutively open channel, whereas substituting this residue with hydrophobic ones significantly shifted the function-voltage curve toward right and reduced the open probability of the channel[35]. These

results are consistent with our aBK simulations with mutations of the corresponding residue Ala305: the Ala305Glu mutant in both the open and closed states showed high ion permeability, whereas Ala305Val and Ala305Leu mutants dehydrated the cavity and prevented ion conduction across the open structure (Fig. 2e).

Non-equilibrium simulations indicated that cavity dewetting is a key step in channel conformational transition from the open to the closed state of both aBK[22] and MthK[21]. Different methods revealed a similar conformational transition mechanism, which involves a rigid body motion of the inner TM helix hinging around a kink in the middle of the helix (Figs. 4e, 5d, e). Cavity wetting-induced channel opening revealed a consistent model. This kink is a key difference between the open and closed states. It is formed due to a glycine that is conserved among most potassium channels[36]. A water molecule hydrogen bonding to the protein backbone is critical for its formation, as indicated by our simulations (Fig. 5). The importance of the glycine and the formation of the kink for channel opening was not only highlighted by our simulations (Fig. 4, Supplementary Figs. 14–17) but also proved by experiments. Mutation of the glycine at corresponding positions in hBK dramatically reduced the open probability of the channel[35]. Note that the conformational transition found by non-equilibrium simulations can also happen spontaneously in equilibrium simulations of MthK[37], confirming the reliability of this work.

Conformational transitions of the four inner helices happened asymmetrically. One possible reason for this observation is the truncation of the protein. Coupling of the TM domain to the calcium-binding domains or the voltage sensing domain is supposed to provide important driving forces for conformational transitions, in addition to cavity wetting/dewetting. However, we do not rule out the possibility of intrinsic asymmetrical conformational transitions of the protein. The timescale of conformational transitions in our simulations was at hundreds of nanoseconds to a few microseconds. This is likely faster than in experiments due to the non-equilibrium techniques used in the current work, although it is possible that the transitions are rare rather than inherently slow. Experiments suggested a sub-millisecond time-scale for conformational transition of KcsA[38,39], while MD simulations of the $K_V1.2/K_V2.1$ "paddle chimera"[29] revealed a spontaneous closure of the pore in ~20 μs and a complete open-to-closed transition in ~100–200 μs.

The fact that non-equilibrium simulations induced a stable closed state of aBK but only a dewetted intermediated state for MthK might indicate the differences between the channels utilizing different mechanisms. For channels in which hydrophobic gating plays a role, a dewetted cavity may be enough to stabilize the closed state, but for channels with bundle crossing, other elements in addition to dewetted cavity may be essential to seal the cavity (for instance, interactions between hydrophobic residues of the inner TM helices[21]).

The most notable difference between the closed states of aBK and MthK is the bending of their inner TM helices. In the cryo-EM structure of the closed state of MthK, it is an almost ideal, straight α helix (Fig. 6a), while in aBK, it bends due to Pro309, which disrupt the backbone hydrogen bonds of α helices (Supplementary Figs. 11, 12). This bending leaves the central cavity open to some extent and prevents the TM helices to intersect at the cytoplasmic side, which may finally result in a different activation gating mechanism in BK channels. However, an elimination of the bending by mutation of Pro309 unlikely re-introduces intersection of the TM helices in our equilibrium and non-equilibrium simulations due to steric hindrance between the TM helices (Fig. 4, Supplementary Figs. 14, 16). We therefore hypothesize minor but important differences in the overall arrangement of the tetramer structure between MthK and BK.

Lipids entering the cavity were found in both the open and closed states of the aBK channel. In the open state, the four TM helices are splayed so much that there is an opening between adjacent subunits facing the hydrophobic region of the bilayer. Lipids at the protein-lipid interface then filled up this opening by their headgroups (Supplementary Fig. 26, Supplementary Note 6) to avoid interactions between water molecules and hydrophobic lipid tails. In most cases, this lipid-protein interaction did not affect the cavity hydration level (Supplementary Table 1). However, structural fluctuations in simulations sometimes moved the adjacent subunits further away from each other and these lipids could enter into the cavity easily (Supplementary Fig. 26), which then induced partial or complete dewetting of the channel, depending on the number of lipid tails inside the cavity. In the closed state, lipid entering into the cavity also needed unusual openings between adjacent subunits and often led to collapse of the tetramer structure.

The physiological relevance of the abovementioned lipid-protein interactions is unclear. We propose that multiple effects, including truncation of the protein in simulations, force field shortcomings, as well as the medium resolution of the cryo-EM structures, may account for this observation. In particular, the fact that lipid entering often induced collapse of the tetramer structure of the closed state renders the lipid-protein interactions perhaps unlikely to be physiologically relevant. However, we do not rule out the possibility that lipid distribution in the cavity is a gating mechanism for potassium channels. Actually, lipid tails entering into the cavity from the fenestration between the four TM helices were found to block the permeation pathway in hBK[25] and MthK[21], as well as for some two pore domain potassium channels[16], or to help to open the pore in Kir channel[34]. Although a similar phenomenon (one of the two lipid tails entering from the fenestration) was found in our simulations, the ratio of the event was very low (Supplementary Fig. 26, 1 out of 12 simulations, in most cases lipid distributions in the cavity induced collapse of the termer structure of the closed state, as mentioned in the Supplementary Note 6). Besides, the following observations might render the idea of lipid-protein interaction as a gating mechanism less likely in our case: (a) >2 lipids in the cavity of the open state dehydrated the cavity completely but did not induce conformational transition from the open to the closed state (see Supplementary Fig. 26b); (b) even without lipid distribution in the cavity, the hydration level of the closed state was low enough to inhibition ion conduction at high potassium concentration (1 M) and voltage (300 mV); (c) in simulations of the closed cryo-EM structure with the inner helices restrained, the cavity was fully hydrated and the lipids could not enter into the cavity. We note that the current work was not meant to sample lipid-protein interaction extensively and further analyses/studies will be required to assess its role in gating of particular potassium channels[16,21,25,28,33].

We also note that the dummy atoms used in our simulations to exclude lipids from the cavity do not affect the main conclusion of this work. Our aim is to investigate the effects of cavity hydration level on ion permeation and the coupling between cavity dewetting and protein conformational transition, excluding lipids allowed sampling different hydration levels with statistical significance. Moreover, the hydration levels and structures of both the open and closed state we sampled with dummy atoms applied are consistent with the simulations in which dummy atoms were not used and lipids were not distributed in the cavity, as mentioned above (Supplementary Table 1).

In summary, we conducted molecular dynamics simulations to investigate the effects of cavity hydration level on the structure and function of potassium channels. Simulations with transmembrane voltage indicated channel activity as a function of cavity hydration level and revealed how a hydrophobic gate modulates ion conduction. Non-equilibrium simulations sampled conformational transition from the open to the closed states of the aBK and MthK potassium channels and proved the coupling between the conformational transition process and cavity dewetting. Our work explored the concept of hydrophobic gating in potassium channels and revealed cavity dewetting as a key step in channel inhibition. The conclusion may also apply to other channels.

*Note added in proof:* While this paper was in production, Nordquist et al. published a paper[40] that examined free energy changes due to (de)wetting transitions in BK channels. Their findings are consistent with those reported here.

## Methods

### Molecular dynamics (MD) simulations

Simulations conducted in this work were summarized in Supplementary Table 1–3. Cryo-EM structures of the open (PDB entry: 5tj6) and closed (PDB entry: 5tji) states of Aplysia BK channel[22] were used to perform molecular dynamics (MD) simulations. In addition to wild type channels, we also conducted simulations of a series of mutants. Ala305, a residue of the inner TM helices whose side chain pointing toward the cavity (Fig. 1), was mutated to Val, Leu, and Glu, respectively, to modulate the cavity hydration level in both the open and closed states. Phe304 and Ile308 were inferred to be responsible for cavity dewetting in the closed state (Fig. 1) and thus were mutated to Ala to control cavity hydration level as well. Gly302Ala and Pro309Ala (see Fig. 1 for their positions) were also used to study the conformation of the closed state and the gating process, as these residues disrupted backbone hydrogen bonds and affected the geometry and tetramer arrangement of the inner TM helices, which constituted the permeation pathway. MD simulations of the crystal structure of MthK[41] (PDB entry: 3ldc), a homolog of BK channels, were also performed to validate the conclusions revealed by BK channel simulations.

We conducted both equilibrium and non-equilibrium MD simulations. The equilibrium simulations employed wild type channel and the abovementioned mutants to compare ion conduction rates of different states of the BK channel, and to investigate the effects of hydration level on ion permeation under applied transmembrane voltage.

The non-equilibrium simulations were performed to explore protein conformational transition from the open to closed state and how it is coupled with cavity dehydration. Two different methods were used for these simulations, both of which started from the open-state structure. One method was designed to study how protein conformation reacts to enforced cavity dehydration, specifically, the chain reaction coordinate (see Supplementary Method and Supplementary Fig. 1 for details) developed by refs. 42, 43 were applied to pull water molecules out of the cavity (i.e., enforced dehydration), in the hope of inducing protein conformation changes. The chain reaction coordinate uses a cylinder to define a region (the central cavity in our case) whose hydration degree is controlled. This cylinder is cut into slices and the ratio of hydrated slices is considered as the reaction coordinate. Forces were applied to the oxygen atoms of all water molecules in the system to pull the water molecules in the cavity slice by slice along the reaction coordinate (see Supplementary Methods for a detailed description). The other method was designed to study changes of cavity hydration level as a result of protein conformational transition. In these simulations, a dummy atom was restrained in the vicinity of Gly302 of the inner TM helices, which formed a kink that is stabilized by a water molecule. The dummy atom had no interactions with the other atoms in the system, except for weak repulsion with the oxygen atoms of waters (see Supplementary Methods). Repelling of waters from the kink prompted helical hydrogen bond re-formation of the kink and a conformation transition from the open to closed state, which may affect hydration level of the central cavity.

We also studied the conformational transition from the closed to the open state induced by cavity hydration. These simulations were started from the cryo-EM structure of the closed state. Two groups of simulations were considered. One was equilibrium simulations of the Ala305Glu mutant, which was also used to analyze ion conduction rate under voltage. The other one was enforced hydration simulations, in which water molecules were restrained in the cavity of the channel by the chain reaction coordinate in order to drive a conformational transition.

In addition, we also performed simulations with restraints applied on backbone hydrogen bonds of the inner TM helices to explore the closed state in details, see Supplementary Table 2.

The simulation systems were constructed by CHARMM-GUI[44]. The systems contained 1 channel, ~160 POPC lipids, ~12,000 water molecules and ~ 220 $K^+$ and similar number of counter ions, constituting a potassium ion concentration of 1 M. MD simulations were conducted by GROMACS2019/2020 package[45], using the CHARMM36m force field[46] and TIP3P water model[47]. A constant electrostatic field (E) of ~0.032 V nm⁻¹ was applied for the equilibrium simulations to mimic a transmembrane voltage (V) of ~300 mV ($V = E \times L$, in which L is the box length along the z-direction). We note that dummy atoms which only have weak repulsions with the carbon atoms from lipid tails were placed in the central cavity to prevent lipids entering into the cavity during simulations (Supplementary Fig. 2, see Supplementary Methods). Each system was equilibrated for 0.25 μs without applying transmembrane voltage before the production simulation, with the protein backbone restrained for the first 0.05 μs and restraints removed for the following 0.2 μs. 5–20 parallel simulations were then conducted for each system with simulation time of a single trajectory ranging from 0.5 to 6 μs. In these production simulations, a transmembrane voltage was applied for the equilibrium simulations, but not for the non-equilibrium simulations which were used to explore protein conformational transitions. For the equilibrium simulations, we also conducted control simulations using different transmembrane voltages and ion concentrations for the wild type and some mutants to further verify our results (1 M/150 mV, 0.15 M/300 mV, and 0.15 M/150 mV, as summarized in Supplementary Table 1). The total simulation time of this work is about ~380 μs. The simulation parameters for the equilibrium and non-equilibrium simulations were described in detail in the Supplementary Methods.

### Data analyses

The methods for the calculation of ion currents, free energy profiles of water molecules/ions entering into the central cavity, number of waters in the central cavity, characterization of the conformation of the inner TM helices, and structural comparison were described in detail in the Supplementary Methods.

### Reporting summary

Further information on research design is available in the Nature Portfolio Reporting Summary linked to this article.

## Data availability

The experimental protein structures used in this work are available from the protein data bank (PDB entry: 5tj6, 5tji, 3ldc, 6u6d). The data generated from MD simulations in this study underlying Figs. 2d-g, 3b, 4b-e, 5b-d, Supplementary Tables 1, 3-4, and Supplementary Figs. 4, 5b, 6, 8, 9, 10a–d, 11, 12b-d, 13, 14b-d, 15, 17, 20, 21b-d, 22b-e, 23d-f, 24, 27, as well as the initial and final conformations of the MD simulations, have been deposited to the Figshare database and are available from https://doi.org/10.6084/m9.figshare.22194646.v1.

## Code availability

Python scripts to calculate orientations of Phe304 sidechains, numbers of water molecules and potassium ions in the cavity, numbers of water molecules in the solvation shells of potassium ions, the free energy profiles of water molecules and potassium ions are available from https://doi.org/10.6084/m9.figshare.22194496.v1.

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

## Acknowledgements

The authors thank Dr. Jochen Hub for sharing a modified version of GROMACS package with the chain reaction coordinate implemented. R.-X.G. is supported by National Natural Science Foundation of China (Grant No. 22107070) and Shanghai Pujiang Program (Grant No. 21PJ1405100). Part of the computations was run on the π2.0 cluster supported by the Center for High Performance Computing at Shanghai Jiao Tong University.

## Author contributions

B.L.d.G. designed and supervised the project. R.-X.G. performed molecular dynamics simulations and analyzed the data. R.-X.G. and B.L.d.G. wrote the manuscript.

## Funding

## Competing interests

The authors declare no competing interests.
