## [Peer Review File · Nature Communications]

Central Cavity Dehydration as A Gating Mechanism of Potassium ChannelsREVIEWER COMMENTS

Reviewer #1 (Remarks to the Author):

In this MD study, the authors examine the interplay between wetting of a hydrophobic gate, ionic conduction, and channel conformation using primarily the K⁺ channel BK as a model system, in which the bundle crossing mechanism of gating is replaced by wetting of the central cavity. In simulations with 1 M KCl and 0.3 V of applied voltage, they find that the ionic current under is proportional to the amount of water in the central cavity of the channel. Using a number of so-called “non-equilibrium” simulations (although all the simulations presented in this manuscript are non-equilibrium), they then explore the relationship between dewetting and channel conformation: initially wetted closed states spontaneously undergo dewetting; enforcing dewetting of the cavity in the open state of the channel leads to conformational collapse of the gate involving a straightening of the kinked pore helix; and finally, forced straightening of the pore helix from the open state leads to dewetting. Mutations confirm the role of hydrophobicity in modulating the hydration of the gate: mutating F and I side chains, which cause dewetting in the wild type, to A abolishes dewetting; and mutating central cavity residue A305 to V, L, and E results in less wetting for V and L and no closure for E.

Overall this is an interesting and well designed, systematic study demonstrating coupling between dewetting and conformational transitions upon closure of the BK channel. The paper is clearly written (although a thorough proofreading is in order) and the results shed light into hydrophobic gating mechanisms in ion channels. However, a limitation of the study is that the reverse wetting and opening transitions are not considered. In addition, controls should be provided to test for possible artefacts of artificially high voltage and ionic concentration. Finally, the results on MthK are underwhelming and could be omitted.

Major issues

Does the relatively large applied voltage influence results qualitatively? Same question for the very large [K⁺] of 1 M. The combination of V and high salt concentration may result in higher ionic occupancy of the central cavity, thereby leading to artificially increased current as well as artificially high hydration. The authors should perform control simulations at physiological [K⁺] to check that their results and conclusions are not affected by the artificially large K⁺ concentration. Would the central cavity be occupied by K⁺ and as many water molecules at physiological [K⁺]? And if not, would that change the results and conclusions?

Why is the conformationally restrained closed state fully wetted (Fig 3)? There are as many as 80 water molecules in the cavity and Fig 2 suggests that this level of hydration corresponds to a fully conducting

channel... This result seems to go against the main conclusions of the paper regarding the role of wetting in ionic conduction.

In turn, this observation raises the issue of possible limitations from only studying open-to-closed transitions in the “non-equilibrium” simulations: if the closed state can be fully wetted, is wetting sufficient to open the channel? While this study examines the coupling between dewetting and conformational gating systematically in channel closure, it does not show that the same principles apply in closed-to-open transitions. The latter could be tested by starting from the (surprisingly) wetted closed channel and forcing water to stay inside the cavity before relaxing conformational restraints on the channel : would the channel relax to the open state? Inversely, other simulations could start with the relaxed, dewetted closed state (except for the kink water) and force the channel to the open conformation to complete the study.

The MthK results are less interesting and convincing than the BK results, first because pulling water out of the cavity was only successful 20% of the time and second, because the channel only reaches a putative intermediate state.

Minor issues

Can you comment on why the current is proportional to the number of water molecules?

Did the channel undergo opening or closing transitions during the simulations? In Fig 2F, the number of water molecules looks like it may be spanning the range of open and closed states.

Where is the central cavity in Fig 2G? Please show the correspondance between z and the channel structure.

Also, please show the whole z range from bulk water to bulk water, including the selectivity filter, which presumably lies on the right beyond the plot.

Why is there a barrier to K⁺ above those at I308 and F304? What is this peak due to ? It is bigger than the barriers discussed in the text in both open and closed states. Does it have an effect on current?

Does the helix kink, by exposing backbone polar groups, also modulate polarity and as such, favor wetting?

Fig 3A: it is hard to distinguish between K⁺ and H₂O, please use different colors.

Please comment on the fact that H bonds between residues 297-301 and 298-302 may not all be formed at once in the 4 helices.

Line 280, the reference to Fig 5 F should be 5E.

Lines 245 and 287 : explain “better” than what?

The effect of lipids, which is only mentioned in passing at the end of the paper (in the Discussion), calls for more details and more references.

In MthK, is wetting correlated to bundle crossing?

Methodological details:

Please clarify whether there is still applied voltage in fig 3-6 results (so-called “non equilibrium” runs).

Why the hot 330K simulations?

Were NBFIXes used for K⁺-channel interactions? How were translocation events counted?

Lines 429-430, references are needed.

Please avoid parallel lists with “respectively” in all figure captions, as they defeat the purpose of captions.

Reviewer #2 (Remarks to the Author):

This is an interesting study on a still important topic, namely the hydrophobic gating model of ion channels. This model has been around for over 2 decades, and now is being subjected to experimental and computational evaluation against a large range of ion channel structures using advanced molecular simulation approaches. In this manuscript computational electrophysiology approaches are used to explore this proposed gating mechanism in potassium channels.

There are two main findings: (i) insufficient hydration is a barrier to ion permeation, as has already been suggested by a number of studies of ion channel structures and models referred to in the manuscript; and (ii) cavity dehydration induces channel conformational changes associated with gating, as had previously been suggested in earlier simulations of K channels (see e.g. Jensen et al., 2012, Science). However, the current study provides a much more extensive study (see Supp Table 1) of hydrophobic gating in potassium channels, using BK and MthK as examples. It employs recent structures and state of the art simulation methods alongside careful and thoughtful analysis, and significantly advances our understanding of ion channel gating in K channels.

For BK channels, an open and a closed state are compared, for ET and mutant structures. Importantly, a strong correlation is seen between the ionic current and the number of waters in the cavity, supporting the hydrophobic gating model. An intermediate level of hydration is seen two mutant channels open/A305V and closed/I308A (I think, the tiny figure 2F makes it difficult to distinguish between the symbols!).

Question: for the 'intermediate' states e.g. open/A305V and closed/I308A the error bars in 2F are larger than for most states/mutants. Does this correspond to stochastic wetting/dewetting of the cavity, which is an important prediction of the hydrophobic gating model. There is some suggestion of this in e.g. Fig. 3B – it would be of interest to see trajectories of individual water molecules perhaps, to help evaluate this.

Also: I note that 'dummy atoms' were used to exclude lipids from the BK cavity (this is rather hidden away in the Supp Information & Supp Fig 2, with a single sentence in the Methods). Is this because just the pore domain of BK is used in the simulations? If so perhaps this should be commented on more explicitly in the main text. I am a bit concerned that the lipid behaviour may be artificially modulated by these restraints, as is hinted at in the section on 'Lipid-protein interactions, Line 418 onwards). This is an important consideration, especially as lipids are included in the e.g. 5TJ6 structure – perhaps these lipids can really enter the pore and modulate gating? This needs to be considered in more detail.

Non-equilibrium simulations: These are used for both BK and MthK channels to 'force' the hydration level of the cavity and these result in some gating-like conformational transitions. Do the authors think the timescales of these conformational transitions match those observed experimentally for K channel gating? This perhaps merits some discussion.

Reviewer #3 (Remarks to the Author):

This manuscript used molecular dynamics simulations of large conductance potassium channels to investigate how pore dewetting influenced ion conduction and also how it alters the conformational state of the protein. While the first aspect is well studied – an effect known as hydrophobic gating, the second is not. The manuscript shows that pore dewetting can induce conformational transitions between open and closed states, and inversely that conformational transitions create pore dewetting. While, this is a bit of a chicken and the egg situation – which causes the other- it does show the two factors are closely linked. The manuscript also shows a very interesting mechanistic role for water molecules in stabilising a kink in the pore lining helices required for opening the pore. Overall, the manuscript is well conducted and provides novel insight.

In the simulations to see how cavity dewetting influenced the conformational state of the channel, water molecules are forced out of the cavity. More details of these forces are required to understand exactly how the process was done (eg if this is done sequentially or the location of the forces etc), not just reference to prior work. How are other water molecules prevented from entering the cavity? More importantly, I was concerned that all water molecules are removed leaving the cavity empty. However, the closed state simulations show that a number of water molecules remain in the cavity. It seems more sensible to me to not completely empty the cavity, but rather to reduce the number of water molecules to that in the closed state and hold it there.

A couple of the important early references for hydrophobic gating have been missed and should be added. These include the work of Rosalind Allen on model pores that pre-dates that of Beckstein and Sansom

A Molecular Dynamics investigation of water permeation through nanopores. R. Allen, J.-P. Hansen and S. Melchionna, *J. Chem. Phys.* 119, 3905 (2003).

Intermittent permeation of cylindrical nanopores by water. R. Allen, S. Melchionna and J.-P. Hansen *Phys. Rev. Lett.* 89, 175502 (2002).

And the following that was possibly the first to show it in a biological pore.

B Corry. An energy efficient gating mechanism in the acetylcholine receptor channel suggested by molecular and Brownian dynamics. *Biophys. J.* 90: 799-810, 2006.

I would have liked to see more analysis of the link between cavity hydration and current. Can you explain why the ion current is reduced so much even when there is still some water in the cavity? Perhaps an analysis of the solvation structure of the ions would help here and provide some more mechanistic insight.

The discussion introduces the fact that lipids sometime penetrate into the cavity, but no data is presented to support this. This is a very important point as it has been suggested that lipid tails in the fenestrations can alter channel gating (or be responsible for gating), either by blocking the pore or by their presence helping open the pore. (<https://www.nature.com/articles/s41467-022-28148-4>) I strongly recommend adding some data and some further analysis her to ensure that the hydration number in the cavity is not being controlled by the presence of lipids.

Reviewer #1 (Remarks to the Author):

In this MD study, the authors examine the interplay between wetting of a hydrophobic gate, ionic conduction, and channel conformation using primarily the K⁺ channel BK as a model system, in which the bundle crossing mechanism of gating is replaced by wetting of the central cavity. In simulations with 1 M KCl and 0.3 V of applied voltage, they find that the ionic current under is proportional to the amount of water in the central cavity of the channel. Using a number of so-called “non-equilibrium” simulations (although all the simulations presented in this manuscript are non-equilibrium), they then explore the relationship between dewetting and channel conformation: initially wetted closed states spontaneously undergo dewetting; enforcing dewetting of the cavity in the open state of the channel leads to conformational collapse of the gate involving a straightening of the kinked pore helix; and finally, forced straightening of the pore helix from the open state leads to dewetting. Mutations confirm the role of hydrophobicity in modulating the hydration of the gate: mutating F and I side chains, which cause dewetting in the wild type, to A abolishes dewetting; and mutating central cavity residue A305 to V, L, and E results in less wetting for V and L and no closure for E.

Overall this is an interesting and well designed, systematic study demonstrating coupling between dewetting and conformational transitions upon closure of the BK channel. The paper is clearly written (although a thorough proofreading is in order) and the results shed light into hydrophobic gating mechanisms in ion channels. However, a limitation of the study is that the reverse wetting and opening transitions are not considered. In addition, controls should be provided to test for possible artefacts of artificially high voltage and ionic concentration. Finally, the results on MthK are underwhelming and could be omitted.

Reply: We thank the reviewer for her/his constructive comments. In this revision, control simulations using lower voltage and potassium concentration were performed to verify our conclusions. We also conducted simulations to study conformational transition from the closed to the open state induced by cavity wetting. Besides, we moved the results of MthK to the supplementary information. Please see the below point to point replies for details.

Major issues

Does the relatively large applied voltage influence results qualitatively? Same question for the very large [K⁺] of 1 M. The combination of V and high salt concentration may result in higher ionic occupancy of the central cavity, thereby leading to artificially increased current as well as artificially high hydration. The authors should perform control simulations at physiological [K⁺] to check that their results and conclusions are not affected by the artificially large K⁺ concentration. Would the central cavity be occupied by K⁺ and as many water molecules at physiological [K⁺]? And if not, would that change the results and conclusions?

Reply: We thank the reviewer for her/his insightful suggestions. In the revision, we conducted control simulations at 1 M/150 mV, 0.15 M/300 mV and 0.15 M/150 mV for some of the simulation systems (both wild type and mutants). The main conclusions were not affected. Although the currents and numbers of potassium ions in the cavity were decreased at lower voltage and ion concentration, similar high correlation between currents and cavity hydration levels were revealed. Also, we did not find statistically significant differences for the numbers of water molecules in the cavity at different voltages and ion concentrations. These new results and discussions were included in page 8 of the main text (second paragraph) and Supplementary Note 1, Supplementary Table 1, and Supplementary Fig. 8 of the supplementary information.

Why is the conformationally restrained closed state fully wetted (Fig 3)? There are as many as 80 water molecules in the cavity and Fig 2 suggests that this level of hydration corresponds to a fully conducting channel... This result seems to go against the main conclusions of the paper regarding the role of wetting in ionic conduction.

Reply: We thank the reviewer for pointing out this important point. As we discussed in Fig. 3, the closed cryo-EM structure was fully wetted because the four TM helices splayed to some extent and the Phe304 sidechains pointed toward subunit interfaces. In this revision, we conducted simulations with the backbone of the inner helices restrained and we found the restrained cryo-EM closed structure was as conductive as the open state at 1 M/300 mV (13.3 v.s. 14.4 pA). Therefore, results in Fig. 3 did not go against the main conclusions but emphasized

the importance of cavity dewetting in channel gating. We mentioned the new results in page 9 of the main text (section “Cavity dewetted quickly in MD simulations of the closed state of BK”).

In turn, this observation raises the issue of possible limitations from only studying open-to-closed transitions in the “non-equilibrium” simulations: if the closed state can be fully wetted, is wetting sufficient to open the channel? While this study examines the coupling between dewetting and conformational gating systematically in channel closure, it does not show that the same principles apply in closed-to-open transitions. The latter could be tested by starting from the (surprisingly) wetted closed channel and forcing water to stay inside the cavity before relaxing conformational restraints on the channel: would the channel relax to the open state? Inversely, other simulations could start with the relaxed, dewetted closed state (except for the kink water) and force the channel to the open conformation to complete the study.

Reply: We thank the reviewer for her/his suggestions and comments. In this revision, we investigated closed-to-open transitions by two groups of simulations: (a) we analyzed protein conformation in the “equilibrium” Ala305Glu mutant simulations using the closed cryo-EM structure as the initial conformation; (b) we also restrained water molecules in the cavity of the closed cryo-EM structure before relax the system, as suggested by the reviewer. We observed hydrated cavities (or partially hydrated cavities at least) and conformational transition toward the open state in both simulations. Although the four subunits behaved asymmetrically and the simulations did not fully converge to a symmetrical open state, these simulations indicated the coupling between cavity hydration and the open state of the channel. We did not conduct simulations starting from the dewetted closed state, but we hope these new results are able to address the reviewer’s concern. These new results were summarized in page 13 of the main text (section “Cavity hydration induced channel opening”) and discussed in detail in Supplementary Note 4 and Supplementary Figs. 21-22.

The MthK results are less interesting and convincing than the BK results, first because pulling water out of the cavity was only successful 20% of the time and second, because the channel only reaches a putative intermediate state.

Reply: We moved the results of MthK to the supplementary information (Supplementary Note 5) and left a brief summary in pages 13-14 of the main text.

Minor issues

Can you comment on why the current is proportional to the number of water molecules?

Reply: We analyzed the numbers of water molecules in the solvation shells of potassium ions (as suggested by another reviewer) and we found the second solvation shells of the ions were all perturbed in the cavity as compared to bulk. The solvation degrees of potassium ions were affected to a different extent, depending on the hydration levels of the cavities, which provided possible explanations for the relationship between currents and the numbers of water molecules in the cavity. These new results and discussions were included in pages 15-16 of the discussion part of the main text and Supplementary Fig. 6 and Supplementary Table 4 in the supplementary information.

Did the channel undergo opening or closing transitions during the simulations? In Fig 2F, the number of water molecules looks like it may be spanning the range of open and closed states.

Reply: We looked at the structures of the mutants in detail and compared them with the closed and open structures of the wild type channel. The Ala305Glu mutant simulations starting from the closed state underwent a conformational transition from the closed to the open state. The Ala305Val and Ala305Leu simulations starting from the open state presented lateral contraction of the cavity to different degrees, rather than a conformational transition (i.e., straightening of the TM helices). Structures of the other mutants maintained their initial backbone structures in the simulations and the changes of the hydration levels were ascribed to the mutated residues. We discussed the effects of mutations on protein structure in the context of current-cavity hydration level relationship in this revision. These results are included in page 7 in the main text (section “Cavity hydration degree correlates with channel function”) and Supplementary Fig. 5 in the supplementary information.

Where is the central cavity in Fig 2G? Please show the correspondance between z and the channel structure.

Reply: We labeled the range of the central cavity in Fig. 2G roughly for clarity and showed the correspondance between z and the channel structures in detail in Supplementary Fig. 4 in the revised version.

Also, please show the whole z range from bulk water to bulk water, including the selectivity filter, which presumably lies on the right beyond the plot.

Reply: We showed the free energy profiles of the whole range in Supplementary Fig. 4 in the revised manuscript.

Why is there a barrier to K⁺ above those at I308 and F304? What is this peak due to? It is bigger than the barriers discussed in the text in both open and closed states. Does it have an effect on current?

Reply: We propose that this barrier was probably due to residues F300 and I301, whose side chains partially pointed toward the cavity and provided a hydrophobic environment together with other residues such as F304 in the cavity, as shown in the correspondance between permeation pathway and the free energy profiles in Supplementary Fig. 4. We hypothesize that they should affect the currents by modulating the distributions of potassium ions there, but avoid to discuss this point in the manuscript as it is out of the scope of this work.

We also took the opportunity to correct a mistake here: the potassium free energy profile of the closed state in our initial manuscript was the results of one simulation replica, we now included the results of all simulations in this revision (Fig. 2G).

Does the helix kink, by exposing backbone polar groups, also modulate polarity and as such, favor wetting?

Reply: We do not think it favors wetting. In the below figure, the surface of the oxygen atoms of residues 297, 298 and the nitrogen atoms of residues 301, 302 are showed in orange. As indicated by the black circle, these atoms mainly faced the subunit interfaces. In this regard, it did not modulate the polarity of the cavity.

Figure 1. Surface of the protein in the open state. Only three subunits are shown for clarify. The oxygen, nitrogen, carbon and sulfur atoms are shown in red, blue, cyan and yellow. The oxygen atoms of residues 297, 298 and the nitrogen atoms of residues 301, 302 are shown in orange.

Fig 3A: it is hard to distinguish between K^+ and H_2O , please use different colors.

Reply: We changed the color of the potassium ions to purple in Fig. 3. We hope they can be discriminated from the water molecules easier.

Please comment on the fact that H bonds between residues 297-301 and 298-302 may not all be formed at once in the 4 helices.

Reply: We ascribed the asymmetrical behaviours of the four TM helices to the truncation of the protein. In *in vivo* systems, coupling between the TM domain and the calcium binding domain should provide important driving forces for conformational transition. However, we do not rule

out the possibility of intrinsic asymmetrical behaviours. We discussed this point briefly in page 17 of the main text (second paragraph) in this revision.

Line 280, the reference to Fig 5 F should be 5E.

Reply: We thank the reviewer for her/his carefully reading. We corrected this mistake in the revision.

Lines 245 and 287: explain “better” than what?

Reply: Line 245: better than the wild type simulations; Line 287: better than simulations without restraining the backbone hydrogen bonds involving Val294. We revised accordingly in pages 11 and 13 in this revision to make the manuscript more readable.

The effect of lipids, which is only mentioned in passing at the end of the paper (in the Discussion), calls for more details and more references.

Reply: In this revision, we (a) described the lipid-protein interactions in detail; (b) compared the results of the simulations using dummy atoms and those without dummy atoms but lipid distributions in the cavity were also not found and concluded that the main conclusions of this work were not affected; (c) discussed the lipid-protein interactions observed in our work against the possibility of lipid-protein interactions as a gating mechanism of the potassium channels in the literatures. These new results and discussions were included in page 14 (section “Lipid-protein interactions”), pages 18-20 of the main text and in Supplementary Note 6 and Supplementary Figs. 26-27 of the supplementary information.

In MthK, is wetting correlated to bundle crossing?

Reply: In MthK, our simulations suggested cavity dewetting was coupled with conformational transition, although the bundle crossing structure was not found. The cavity of the bundle

crossing structure was dewetted in equilibrium simulations, so we assumed that the wetting was not correlated to bundle crossing.

Methodological details:

Please clarify whether there is still applied voltage in fig 3-6 results (so-called “non equilibrium” runs).

Reply: Voltage was not applied in the results of Figs. 3-6. We clarified this point in the method section in page 22 of the main text and in Supplementary Table 1.

Why the hot 330K simulations?

Reply: The higher temperature was used for enhanced sampling. We added a note to clarify this point in Supplementary Table 1.

Were NBFIXes used for K⁺-channel interactions? How were translocation events counted?

Reply: We did not use NBFIXes in the current simulations, based on the results from this publication <https://pubs.acs.org/doi/10.1021/ja103270w>. A potassium ion translocated from the S_c binding site to the S₀ binding site was counted as a permeation event. The counting was realized by a Fortran script, which was shared in our previous publications (Kopeck et al., *Nat. Commun.*, 2019). We clarified this point in page 11 (section “Ion permeability”) of the Supplementary Method.

Lines 429-430, references are needed.

Reply: We added references at corresponding positions of the revision (the end of the first paragraph in page 19)

Please avoid parallel lists with “respectively” in all figure captions, as they defeat the purpose of captions.

Reply: We revised the figure captions accordingly in the revised manuscript.

Reviewer #2 (Remarks to the Author):

This is an interesting study on a still important topic, namely the hydrophobic gating model of ion channels. This model has been around for over 2 decades, and now is being subjected to experimental and computational evaluation against a large range of ion channel structures using advanced molecular simulation approaches. In this manuscript computational electrophysiology approaches are used to explore this proposed gating mechanism in potassium channels.

There are two main findings: (i) insufficient hydration is a barrier to ion permeation, as has already been suggested by a number of studies of ion channel structures and models referred to in the manuscript; and (ii) cavity dehydration induces channel conformational changes associated with gating, as had previously been suggested in earlier simulations of K channels (see e.g. Jensen et al., 2012, Science). However, the current study provides a much more extensive study (see Supp Table 1) of hydrophobic gating in potassium channels, using BK and MthK as examples. It employs recent structures and state of the art simulation methods alongside careful and thoughtful analysis, and significantly advances our understanding of ion channel gating in K channels.

For BK channels, an open and a closed state are compared, for ET and mutant structures. Importantly, a strong correlation is seen between the ionic current and the number of waters in the cavity, supporting the hydrophobic gating model. An intermediate level of hydration is seen two mutant channels open/A305V and closed/I308A (I think, the tiny figure 2F makes it difficult to distinguish between the symbols!).

Reply: We thank the reviewer for her/his positive comments. The symbols in Fig. 2F were enlarged to improve the readability in the revision.

Question: for the ‘intermediate’ states e.g. open/A305V and closed/I308A the error bars in 2F are larger than for most states/mutants. Does this correspond to stochastic wetting/dewetting of the cavity, which is an important prediction of the hydrophobic gating model. There is some suggestion of this in e.g. Fig. 3B – it would be of interest to see trajectories of individual water molecules perhaps, to help evaluate this.

Reply: The larger error bars of open/A305V and closed/I308A were due to structural fluctuations of the proteins in different simulation replicas, which affected the numbers of water molecules in the cavity. To address the reviewer’s concern regarding the stochastic wetting/dewetting of the channel, we analyzed the fluctuations of the numbers of water molecules in the cavities using the relative standard deviations (RSD, ratio of the standard deviation to the mean). We found larger fluctuations for the more dewetted cavities. We also took the simulations of the closed wild type channel (the most dewetted case) as an example to look at trajectories of individual waters to show the wetting/dewetting details. The results were summarized briefly in page 8 of the main text (the third paragraph) and Supplementary Fig. 10. However, we avoid further discussions in the context of stochastic wetting/dewetting to avoid overinterpretation of the results.

Also: I note that ‘dummy atoms’ were used to exclude lipids from the BK cavity (this is rather hidden away in the Supp Information & Supp Fig 2, with a single sentence in the Methods). Is this because just the pore domain of BK is used in the simulations? If so perhaps this should be commented on more explicitly in the main text. I am a bit concerned that the lipid behaviour may be artificially modulated by these restraints, as is hinted at in the section on ‘Lipid-protein interactions, Line 418 onwards). This is an important consideration, especially as lipids are included in the e.g. 5TJ6 structure – perhaps these lipids can really enter the pore and modulate gating? This needs to be considered in more detail.

Reply: We described and discussed the lipid-protein interactions in more detail in this revision. We do not rule out the possibility that protein truncation is responsible for observed lipid

distributions in the cavity. We also do not rule out the possibility that lipid distributions really work as a gating mechanism. We discussed these possibilities in the revised manuscript.

However, lipid-protein interactions did not happen in all simulations and showed different interaction modes in the simulation replicas they entered into the cavity. Moreover, they often led to collapse of the tetramer structure in the closed state. In this regard, excluding lipids out of the cavity allowed us to maintain the tetramer structure of the channel and to sample different cavity hydration levels with statistical significance. Besides, comparison of the simulations using dummy atoms to simulations without using dummy atoms but lipids distributions in the cavity were not found showed consistent results, suggesting the main conclusions were not affected. These new results and discussions were included in page 14 (section “Lipid-protein interactions”), pages 18-20 of the main text and in Supplementary Note 6 and Supplementary Figs. 26-27 of the supplementary information.

Non-equilibrium simulations: These are used for both BK and MthK channels to ‘force’ the hydration level of the cavity and these result in some gating-like conformational transitions. Do the authors think the timescales of these conformational transitions match those observed experimentally for K channel gating? This perhaps merits some discussion.

Reply: The timescale of conformational transitions in our non-equilibrium simulations was at hundreds of nanoseconds to a few microseconds. This is possibly faster than in experiments, which revealed a timescale between sub-millisecond to milliseconds, although the process itself may be rare rather than inherently slow. We discussed this point in page 17 of the discussion part (the second paragraph) in the main text.

Reviewer #3 (Remarks to the Author):

This manuscript used molecular dynamics simulations of large conductance potassium channels to investigate how pore dewetting influenced ion conduction and also how it alters the conformational state of the protein. While the first aspect is well studied – an effect known as hydrophobic gating, the second is not. The manuscript shows that pore dewetting can induce

conformational transitions between open and closed states, and inversely that conformational transitions create pore dewetting. While, this is a bit of a chicken and the egg situation – which causes the other- it does show the two factors are closely linked. The manuscript also shows a very interesting mechanistic role for water molecules in stabilising a kink in the pore lining helices required for opening the pore. Overall, the manuscript is well conducted and provides novel insight.

In the simulations to see how cavity dewetting influenced the conformational state of the channel, water molecules are forced out of the cavity. More details of these forces are required to understand exactly how the process was done (eg if this is done sequentially or the location of the forces etc), not just reference to prior work. How are other water molecules prevented from entering the cavity? More importantly, I was concerned that all water molecules are removed leaving the cavity empty. However, the closed state simulations show that a number of water molecules remain in the cavity. It seems more sensible to me to not completely empty the cavity, but rather to reduce the number of water molecules to that in the closed state and hold it there.

Reply: We thank the reviewer for her/his insightful comments. In the reaction coordinate, the forces are applied on the oxygen atoms of all water molecules in the system to pull the water molecules in the cavity slice by slice (as it uses the ratio of hydrated slices as the reaction coordinate). Specifically, a harmonic potential of $V = k(\xi - \xi_0)^2 / 2$ is applied to the system (ξ_0 is the reference point along the reaction coordinate). The force applied to the system is calculated as $F_\xi = -\partial V / \partial \xi = -k(\xi - \xi_0)$, which was then translated to the atoms by $F_j = F_\xi \cdot \partial \xi / \partial r_j$ (j is the atom index). We summarized this in page 21 of the method section in the main text and described in more detail in page 10 of the Supplementary Method.

It is really a good idea to just reduce the number of water molecules rather than empty the cavity. However, as aforementioned, the chain reaction coordinate used in this work pulled water molecules slice by slice. In this regard, the water molecules will be restrained at some slices rather than distributed evenly in the cavity if we leave a number of water molecules in the cavity, which may not what we want. Therefore, we did not conduct the suggested simulations in this revision.

However, we argue that some water molecules entered into the cavity after the restraints were removed and the cavity was hydrated to a level comparable to the equilibrium simulations of the closed state (Fig. 4C). This observation may rationalize the structures we sampled.

A couple of the important early references for hydrophobic gating have been missed and should be added. These include the work of Rosalind Allen on model pores that pre-dates that of Beckstein and Sansom

A Molecular Dynamics investigation of water permeation through nanopores. R. Allen, J.-P. Hansen and S. Melchionna, *J. Chem. Phys.* 119, 3905 (2003).

Intermittent permeation of cylindrical nanopores by water. R. Allen, S. Melchionna and J.-P. Hansen *Phys. Rev. Lett.* 89, 175502 (2002).

And the following that was possibly the first to show it in a biological pore.

B Corry. An energy efficient gating mechanism in the acetylcholine receptor channel suggested by molecular and Brownian dynamics. *Biophys. J.* 90: 799-810, 2006.

Reply: We thank the reviewer for pointing out these important papers. We have cited and discussed them in the first paragraph of the introduction section in the revised manuscript.

I would have liked to see more analysis of the link between cavity hydration and current. Can you explain why the ion current is reduced so much even when there is still some water in the cavity? Perhaps an analysis of the solvation structure of the ions would help here and provide some more mechanistic insight.

Reply: We thank the reviewer for this constructive suggestion. We evaluated the numbers of water molecules in the solvation shells of potassium ions. The average numbers and distributions suggested that the second solvation shells were perturbed for the potassium ions in the cavities of all simulation systems compared to the bulk. The solvation shells were affected to a different extent, depending on the hydration levels of the cavity. This observation may provide some

insight into the current-cavity hydration level relationship. These new results and discussions were included in pages 15-16 of the discussion part of the main text and Supplementary Fig. 6 and Supplementary Table 4 in the supplementary information.

The discussion introduces the fact that lipids sometime penetrate into the cavity, but no data is presented to support this. This is a very important point as it has been suggested that lipid tails in the fenestrations can alter channel gating (or be responsible for gating), either by blocking the pore or by their presence helping open the pore. (<https://www.nature.com/articles/s41467-022-28148-4>) I strongly recommend adding some data and some further analysis here to ensure that the hydration number in the cavity is not being controlled by the presence of lipids.

Reply: We thank the reviewer for her/his constructive suggestions. As mentioned in the replies to another reviewer's comments, in this revision, we (a) described the lipid-protein interactions in detail; (b) compared the results of the simulations using dummy atoms and those without dummy atoms but lipid distributions in the cavity were also not found and concluded that the main conclusions of this work were not affected; (c) discussed the lipid-protein interactions observed in our work against the possibility of lipid-protein interactions as a gating mechanism of the potassium channels in the literatures.

We note that lipid distributions in the cavity were not possible to be sampled with statistical significance in accessible simulation time, as they did not enter into the cavity in all simulation replicas and they showed different interaction modes in the cases they entered. Also, lipid entering into the cavity often led to collapse of the tetramer structure. In this regard, excluding them out of the cavity allowed us to sample different hydration levels and maintain the tetramer structures of the channel.

The abovementioned results and discussions were included in page 14 (section "Lipid-protein interactions"), pages 18-20 of the main text and in Supplementary Note 6 and Supplementary Figs. 26-27 of the supplementary information.

REVIEWERS' COMMENTS

Reviewer #1 (Remarks to the Author):

The response and revision include a significant number of additional simulations and analyses that address most of my (and other reviewers') previous concerns satisfactorily.

The manuscript would benefit from the following minor clarifications and corrections:

It is puzzling that the pore of the channel in the EM structure of the presumably closed state is large enough to be as hydrated and as conductive as the open state. While the resolution of the EM data is likely too low to assign the conformation of the Phe side chains correctly, the "splaying" of the pore helices in the EM structure of the closed state, as mentioned by the authors in their response to my query, is still surprising. While explaining this feature is of course outside the scope of this paper, I do think that it would help the readers if the authors would echo their response with a brief comment highlighting this apparent paradox in the text.

Since the spontaneous structural relaxation of the "presumably closed" EM structure from a wetted to a dewetted state was also observed in the previous MD study of MthK by Jia et al. (ref. 25 in the manuscript), it would be appropriate to mention that corroborating precedent in the text.

In the same line of thought, it would help to specify what is meant by "closed state" to avoid confusing the wetted (restrained) EM structure and the computationally relaxed, dewetted state.

Please specify that the data in Supplementary Figure 8 corresponds to non-restrained simulations.

On Line 347, please change "to the open state" —> "towards the open state" since the open state has not been reached.

In Supplementary Note 4, please explain how the structure was "restrained to be hydrated" (methodology) for the sake of reproducibility.

Please provide missing units in Fig. S21.

In Fig. S22, please clarify what you mean by “pulled”.

In Supplementary Fig. 6, labeling the y-axis as “percentage” is misleading since you are not showing a histogram but a continuous curve, so that the scale is arbitrary.

For the sake of accuracy, please refer to “the first two solvation shells” rather than “the second solvation shell” of the ions whenever appropriate (e.g. line 422).

The response to our previous question regarding bundle crossing is unclear. Please clarify.

Reviewer #2 (Remarks to the Author):

The revisions to the manuscript have answered all the points I raised earlier, and more generally have improved the manuscript. I am now entirely happy for this to proceed.

Reviewer #3 (Remarks to the Author):

The authors have done an excellent job responding to the initial round of reviews. I believe that they have satisfactorily incorporated all my suggestions and addressed by concerns.

This is a solid manuscript and I believe it should be published.

Reviewer #1 (Remarks to the Author):

The response and revision include a significant number of additional simulations and analyses that address most of my (and other reviewers') previous concerns satisfactorily.

The manuscript would benefit from the following minor clarifications and corrections:

It is puzzling that the pore of the channel in the EM structure of the presumably closed state is large enough to be as hydrated and as conductive as the open state. While the resolution of the EM data is likely too low to assign the conformation of the Phe side chains correctly, the “splaying” of the pore helices in the EM structure of the closed state, as mentioned by the authors in their response to my query, is still surprising. While explaining this feature is of course outside the scope of this paper, I do think that it would help the readers if the authors would echo their response with a brief comment highlighting this apparent paradox in the text.

Reply: We thank the reviewer for her/his suggestions. We have highlighted this paradox at the third paragraph in page 9 of the main text in this revision.

Since the spontaneous structural relaxation of the “presumably closed” EM structure from a wetted to a dewetted state was also observed in the previous MD study of MthK by Jia et al. (ref. 25 in the manuscript), it would be appropriate to mention that corroborating precedent in the text.

Reply: We thank the reviewer for her/his constructive suggestions. However, we have already discussed Jia et al.'s work to support our observations at the end of the second paragraph in page 9 of the main text.

In the same line of thought, it would help to specify what is meant by “closed state” to avoid confusing the wetted (restrained) EM structure and the computationally relaxed, dewetted state.

Reply: We added a note in page 9 of the main text to clarify that the “closed state” refers to the relaxed conformation with dewetted cavity.

Please specify that the data in Supplementary Figure 8 corresponds to non-restrained simulations.

Reply: We thank the reviewer for her/his suggestions. We specified this point at two positions: (1) we noted in Supplementary Table 1 as “no restraints applied” for the simulations in which we did not apply any restraints for the protein; (2) we also indicated in the legend of Supplementary Figure 8 that all of the simulations mentioned in this figure did not apply any restraints on proteins.

On Line 347, please change “to the open state” —> “towards the open state” since the open state has not been reached.

Reply: We have revised accordingly in page 13 of the main text.

In Supplementary Note 4, please explain how the structure was “restrained to be hydrated” (methodology) for the sake of reproducibility.

Reply: We thank the reviewer for her/his constructive suggestions. The method for restraining was already described in detail in the Supplementary Methods section in pages 10-11 of the Supplementary Information. In the revision, we indicated that this restraint was applied by the chain reaction coordinate and referred the Supplementary Methods for the readers for details (page 5 of the Supplementary Information).

Please provide missing units in Fig. S21.

Reply: We thank the reviewer for her/his suggestions. We included the units of the “RMSD” and the “298O-302HN distance” in panel C of the Supplementary Figure S21. Note that the units are provided in the figures, instead of the labels of the y-axes, in order to avoid very crowded labels.

In Fig. S22, please clarify what you mean by “pulled”.

Reply: The “pulled” simulations referred to the non-equilibrium simulations of the wild type channel (as the water molecules were pulled). We clarified this point in the legend of Supplementary Figure 22.

In Supplementary Fig. 6, labeling the y-axis as “percentage” is misleading since you are not showing a histogram but a continuous curve, so that the scale is arbitrary.

Reply: We labelled the y-axis of Supplementary Figure 6 as “Probability” in the revision.

For the sake of accuracy, please refer to “the first two solvation shells” rather than “the second solvation shell” of the ions whenever appropriate (e.g. line 422).

Reply: We thank the reviewer for her/his suggestions. We have revised accordingly in pages 15-16 of the main text and in the title of Supplementary Table 4 and the legend of Supplementary Figure 6.

The response to our previous question regarding bundle crossing is unclear. Please clarify.

Reply: In the reviewer’s previous comments, she/he asked “In MthK, is wetting correlated to bundle crossing? ”.

When the water molecules are pulled out of the cavity, the structure of MthK changed from the open state to an intermediate state in which the cavity was dewetted (with ~10 water molecules, see Supplementary Figure 23) but the TM helices still bent to some extent. Although this structure is able to inhibit ion permeation as suggested by our previous work (Gu et al., Nat. Commun., 2020), it is different from the bundle crossing conformation in that, the TM helices are “straight” in the bundle crossing structure. We are sampling the conformational transition from this intermediate state to the bundle crossing conformation in an ongoing work, which

shows that bundling crossing of the TM helices further drive the remaining water molecules out of the cavity. Based on all of these simulations, we conclude that cavity dewetting is a key step in the conformational transition from the open state to the bundle crossing state in MthK.